# Design and Analysis of a Single System of Impedimetric Biosensors for the Detection of Mosquito-Borne Viruses

**DOI:** 10.3390/bios11100376

**Published:** 2021-10-07

**Authors:** Fahmida Nasrin, Kenta Tsuruga, Doddy Irawan Setyo Utomo, Ankan Dutta Chowdhury, Enoch Y. Park

**Affiliations:** 1Laboratory of Biotechnology, Research Institute of Green Science and Technology, Shizuoka University, 836 Ohya, Suruga-ku, Shizuoka 422-8529, Japan; fahmida.nasrin.17@shizuoka.ac.jp (F.N.); ankan.dutta.chowdhury@shizuoka.ac.jp (A.D.C.); 2Laboratory of Biotechnology, Department of Agriculture, Graduate School of Integrated Science and Technology, Shizuoka University, 836 Ohya, Suruga-ku, Shizuoka 422-8529, Japan; tsuruga.kenta.17@shizuoka.ac.jp; 3Laboratory of Biotechnology, Graduate School of Science and Technology, Shizuoka University, 836 Ohya, Suruga-ku, Shizuoka 422-8529, Japan; doddy.irawan.setyo.utomo.16@shizuoka.ac.jp

**Keywords:** dengue virus, dengue serotype, mosquito-borne viral disease, virus detection, electrochemical impedance spectroscopy

## Abstract

The treatment for mosquito-borne viral diseases such as dengue virus (DENV), zika virus (ZIKV), and chikungunya virus (CHIKV) has become difficult due to delayed diagnosis processes. In addition, sharing the same transmission media and similar symptoms at the early stage of infection of these diseases has become more critical for early diagnosis. To overcome this, a common platform that can identify the virus with high sensitivity and selectivity, even for the different serotypes, is in high demand. In this study, we have attempted an electrochemical impedimetric method to detect the ZIKV, DENV, and CHIKV using their corresponding antibody-conjugated sensor electrodes. The significance of this method is emphasized on the fabrication of a common matrix of gold–polyaniline and sulfur, nitrogen-doped graphene quantum dot nanocomposites (Au-PAni-N,S-GQDs), which have a strong impedimetric response based only on the conjugated antibody, resulting in minimum cross-reactivity for the detection of various mosquito-borne viruses, separately. As a result, four serotypes of DENV and ZIKV, and CHIKV have been detected successfully with an LOD of femtogram mL^−1^.

## 1. Introduction

The severity of mosquito-borne diseases is a global threat caused by protozoa, viruses, or parasites, resulting in nearly 700 million illnesses and over one million deaths each year [1]. The annual epidemic for protozoa-caused malaria has been a deadly problem in tropical regions since the nineteenth century. However, in recent decades, the outbreaks of diseases such as dengue, zika, and chikungunya viruses are widely epidemic throughout the summer and rainy season [2,3,4]. Zika virus (ZIKV), dengue virus (DENV), and chikungunya virus (CHIKV) are vector-borne human viral pathogens, sharing the same vectors of Aedes aegypti or Aedes albopictus [5,6]. The fatality of these viral diseases is significantly high, especially in highly populated regions due to their fast transmission rates and delays in initial diagnosis. Symptomatic diagnosis is more critical because ZIKV, DENV, and CHIKV share the same transmission media and similar clinical manifestations such as fever, myalgia, and headaches at the early stage of infection [7,8,9]. In 2019, over 5.2 million dengue cases were reported worldwide: most significantly in tropical and sub-tropical regions such as India, Bangladesh, and Brazil [10]. Chikungunya and zika have exhibited equivalent fatality over the past few decades in India, Brazil, Bangladesh, Indonesia, etc. Meanwhile, most notably has been that imported cases of ZIKV infection from South America and Oceania were reported in some areas of China, where outbreaks have never previously been reported [11,12]. It indicates that the high spreading ability of these viruses can cause a pandemic if not diagnosed at early stages of infection. 

Among the various diagnosis methods, virus isolation, enzyme-linked immunosorbent assay, and reverse-transcriptase polymerase chain reaction (RT-PCR) have generally been used for detecting viral infections [13,14,15]. Although virus isolation in susceptible cell lines is a highly reliable detection method, it is not an appropriate clinical diagnostic assay for the detection of early infection of these viruses [16,17]. Due to the low level of immunoglobulin M in the early stage and the high possibility of cross-reactivity among these viruses and their subtypes, enzyme-linked immunosorbent assays are also an insufficient method for early diagnosis [18]. Despite some development of nanotechnology-based rapid detection methods, the reliability could not perform adequately close to the gold standard RT-PCR methods [19,20,21,22]. However, all these methods have some limitations in replacing the expensive but standard methods such as RT-PCR. For the diagnosis of multiple viruses with typical symptoms, these methods are extremely time-consuming.

Following the advancement of nanotechnologies in virus detection, many reports can be found for direct or indirect ZIKV, CHIKV, and DENV detection [8,23]. Among the different methods for nanomaterial-based biosensors, fluorometric and electrochemical detection methods have emerged recently due to their simple techniques, fast responses, and cost-effectivity [24,25]. In terms of sensitivity, electrochemical methods are always preferable; however, applications for closely related virus samples are rarely reported and need to be studied more thoroughly.

Inspired by our few recent reports on electrochemical sensing [20,26,27,28,29], in this study we have developed an electrochemical impedimetric biosensor using gold–polyaniline nanocomposites (Au-PAni) and nitrogen, sulfur co-doped graphene quantum dots (N,S-GQDs) as the base matrix [30,31]. EIS is very popular in energy storage, battery, and solid-state electrolyte applications [32,33,34], although applications in biosensing devices are also emerging. In this study, the nanocomposites of Au-PAni and N,S-GQDs were conjugated together with different antibodies and thereafter applied for their corresponding target viruses by the impedimetric process. Plenty of carboxylic groups on GQDs can covalently be attached with the free amino group of antibodies, which makes the electrode surface stable and specific for detection. In addition, due to minimum interactions of the Au-PAni-N,S-GQD towards biological substances, including the target virus, the sensor’s specificity is solely dependent on the antibody–antigen interaction. Therefore, Au-PAni-N,S-GQD nanocomposites have been used in this study to detect different mosquito-transmitted viruses such as CHIKV, ZIKV, and DENV, altering only the antibody on the sensor surface. Successful results with minimal cross-reactivity encouraged us to proceed with a more intense study on DENV serotype detection. Due to proper optimization of the sensor development and blocking of the electrode surface, the sensor showed good responses in the electrochemical impedimetric signal towards the corresponding viruses, even in different serotypes. In all cases, the limit of detection was found to be as low as femtogram mL^−1^ concentration, which confirms the applicability of this sensor for sensitive and rapid detection. The successful results in this study encourage the extension of this research to explore the sensor performance in multiple detection platforms for different mosquito-borne viruses in single-pot measurements in future.

## 2. Materials and Methods

### 2.1. Reagents and Biomaterials

Sodium acetate, sulfuric acid, phosphate-buffered saline (PBS) buffer, aniline, toluene, potassium hydroxide (KOH), hydrochloric acid (HCL), methanol, ethanol, citric acid anhydrous, thiourea, and acetone were purchased from Wako Pure Chemical (Osaka, Japan). Bovine serum albumin (BSA), HAuCl_4_, N-(3-dimethyl aminopropyl)-N′-ethyl carbodiimide hydrochloride (EDC), and N-hydroxysuccinimide (NHS) were purchased from Sigma Aldrich Co., LLC (Saint Louis, MO, USA). Chikungunya virus lysate (strain: Chikungunya virus), Zika virus lysate (strain: Zika virus, ref 1308258v), Mouse anti-Chikungunya virus capsid protein (clone: CA980), and Mouse anti-Zika virus antibody (Clone: ID5-2-H7-G3) were purchased from The Native Antigen Company (Oxfordshire, UK). Bm5 cells were provided by Prof. K. S. Boo (Insect Pathology Laboratory, School of Agricultural Biotechnology, Seoul National University, Seoul, South Korea) and maintained at 27 °C in Sf-900II serum-free medium (Thermo Fisher Scientific K.K., Tokyo, Japan) supplemented with 1% fetal bovine serum and Antibiotic–Antimycotic solution (Thermo Fisher Scientific K.K., Tokyo, Japan).

The monoclonal anti-dengue virus envelope protein of serotype 1 (Clone E29), serotype 2 (Clone 3H5-1), serotype (Clone E1), and serotype 4 (Clone E42) were purchased from Bei Resources (Manassas, VA, USA). For the selectivity test, influenza virus A/H1N1 (New Caledonia/20/99) was purchased from Prospec-Tany Techno Gene Ltd. (Rehovot, Israel), and norovirus-like particle (NoV-LP) was provided by Dr. Tian-Cheng Li (National Institute of Infectious Diseases, Tokyo, Japan).

### 2.2. Equipment

Transmission electron microscopy (TEM) images of nanocomposites were obtained using a TEM system (JEM-2100F; JEOL, Ltd., Tokyo, Japan) operated at 100 kV. Dynamic light scattering (DLS) measurements were performed using a Zetasizer Nano series (Malvern Inst. Ltd., Malvern, UK). Electrochemical impedance spectroscopy (EIS) and electrochemical cyclic voltammetry (CV) were performed by using an SP-150 (BioLogic Inc., Tokyo, Japan), which consists of a conventional three-electrode cell containing platinum wire. Saturated Ag/AgCl was used as an electrolyzer (EC frontier, Tokyo, Japan).

### 2.3. Preparation of the AuNP-PAni Nanocomposite

AuNP-PAni nanocomposites were synthesized via interfacial self-oxidation–reduction polymerization with HAuCl_4_ as an aqueous oxidant and the polyaniline as a monomer in the organic toluene layer [35]. These two immiscible layers made contact at an interface; then, the Au^3+^ ions oxidized the aniline monomer to its conducting emeraldine salt polymer formation in the nanotube structure, whereas it was reduced to the nano Au^0^ form itself. The AuNP was therefore entrapped on the nanotube surface of the polyaniline, resulting in AuNP-PAni nanocomposites. Finally, the AuNP-PAni nanocomposites were drop-casted on the PAni-coated Au electrode for further analysis.

### 2.4. Synthesis of N,S-GQD, and Conjugation with Antibody 

The synthesis of N,S-GQDs was followed by a hydrothermal reaction system [36]. Antibodies for the target virus were bound with N,S-GQDs using EDC/NHS covalent chemistry [37]. Briefly, 0.1 M EDC was mixed with 5.1 µg of antibody solution and reacted with the carboxyl group contained in the Ab after 30 min of stirring at 7 °C. After that, 1 mL of N,S-GQDs, and 0.1 M NHS were added to activate the amino group on the surface of the GQDs and then stirred for 16 h at 7 °C. The reaction solution was dialyzed by using a 1 kDa dialysis bag to remove the excess EDC and NHS. Finally, the antibody-conjugated N,S-GQD (Ab-N,S-GQDs) solution was stored in 0.1 M PBS (pH 7.4) at 4 °C.

### 2.5. Fabrication of the Gold Electrode

Deposition of nanocomposites on the gold electrode produces high conductivity to perform electrochemical analysis. Initially, the polyaniline was electrochemically deposited on the gold surface using cyclic voltammetry (CV) in a three-electrode system to prepare the gold electrode. The obtained curve for CV was recorded at a scan rate of 20 mV/s in a potential range of 0–1 V for 10 cycles. After that, 15 µL of the mixed solution of Ab-N,S-GQD with AuNP-PAni was deposited by drop-casting onto the gold surface with polymerized PAni. The sulfur on N,S-GQD formed strong bonds of Au–S with AuNPs via the soft acid–soft base interaction. To minimize the reactivity of the base matrix of Au-PAni, the final sensor electrode was immersed in a solution of 0.2% BSA for blocking before using it for virus detection.

### 2.6. Preparation of Dengue Virus-Like Particles

The dengue virus-like particles (DENV-LPs) serotypes 1–4 were prepared according to a previously reported protocol [38]. Briefly, the DENV-LPs were expressed in Bm5 cells and purified using affinity chromatography. Transmission electron microscopy revealed that these DENV-LPs formed rough, spherical forms, with a diameter of 30–55 nm. Furthermore, the heparin-binding assay demonstrated that these DENV-LPs contained the envelope protein domain III on their surfaces [39].

### 2.7. Electrochemical Detection of Virus

The virus solution was diluted in a series of concentrations from 10 fg mL^−1^ to 1 ng mL^−1^ using filtered 0.1 M PBS solution. For detection, 10 µL of virus solution was dropped on the gold electrode containing the Ab-N,S-GQD@AuNP-PAni nanocomposite and incubated for 10 min at room temperature. The virus was bound with the antibody-conjugated surface of the electrode. The unbound virus was washed by dipping the electrode in PBS and then kept in the electrolytic solution. The value for charge transfer resistance (*R*_ct_) on the electrode was then measured by the potential EIS with a sinusoidal amplitude of 5 mV within a frequency range from 100 kHz to 0.1 Hz. The virus detection time using the gold electrode was about 15 min.

## 3. Results

### 3.1. Preparation of Sensor Electrode and Its Sensing Mechanism

AuNP-PAni nanocomposites were synthesized via interfacial self-oxidation–reduction polymerization with HAuCl_4_ as an aqueous oxidant and the polyaniline as a monomer in the organic toluene layer. These two layers met at their interfaces; then, the aniline was oxidized to its conducting emeraldine salt polymer formation in the nanotube structure. Au^3+^ was reduced to Au^0^, entrapped on the nanotube surface. The synthesis and their TEM images are shown in Appendix A. To synthesize the sensor electrode with the AuNP-PAni nanocomposites, the bare Au electrode was coated with a fine layer of polyaniline via cyclic voltammetry (Appendix A), as presented in the scheme in Figure 1a. The homogeneously distributed N,S-GQDs were prepared by the standard hydrothermal route and conjugated with monoclonal antibodies (Ab) via the EDC/NHS mechanism. The TEM image of the as-synthesized N,S-GQDs are given in Appendix A. Then, the Ab-conjugated N,S-GQD (N,S-GQD-Ab) was dialyzed overnight and drop-casted on the Au|PAni|Au-PAni electrode to synthesize the sensor electrode Au|PAni|Au-PAni|N,S-GQD-Ab. The conjugation between the sulfur atom of N,S-GQD and the AuNP of AuPAni was made by the universal gold–thiol interaction [40,41].

It can be anticipated that the conductivity and the charge storage property of the sensor electrode should possess a high value due to the presence of a conducting surface of AuNP and PAni nanotubes. As shown in the cyclic voltammogram (Figure 1b), the Au|PAni|Au-PAni electrode had a significantly high charge storage capacity with a clear redox peak at +0.45/+0.68 V due to the most electroactive form of the emeraldine salt of polyaniline, compared to the bare Au and Au|PAni electrode [42]. Similarly, its impedance spectrum also exhibits small resistivity in the Nyquist plot in Figure 1c.

### 3.2. Detection of CHIKV by Au|PAni|Au-PAni-N,S-GQD-Ab_CHIKV_ Sensor Electrode

The detection ability was investigated on chikungunya virus with the Au|PAni|Au-PAni-N,S-GQD electrode with anti-chikungunya antibody (Ab_CHIKV_). The antibody was conjugated with the N,S-GQDs and dialyzed well before proceeding to Au-PAni-N,S-GQD formation. After that, only sensor electrodes without virus loading were recorded in impedance as the control, which was compared with different virus concentrations loaded on the electrodes. The Nyquist plots of the Au|PAni|Au-PAni-N,S-GQD electrode before and after virus loading are shown in Figure 2a, where *Z*′ and *Z*′′ represent the real and imaginary parts, respectively, of the impedance over the frequency range 100 kHz to 100 MHz with an AC amplitude of 5 mV. The plot clearly shows that the increasing pattern in impedance occurred with the increasing virus concentrations from 100 fg·mL^−1^ to 1 ng·mL^−1^. To observe the individual contributions of impedance value, all Nyquist plots were fitted with several possible equivalent circuit diagrams over frequencies ranging from 100 kHz to 100 MHz. The best-fitted diagram was applied to decipher the individual contribution of the circuit parameters, which is depicted in Appendix A. The most crucial factor of these, *R*_ct_, represents the transfer of electrons at the electrode to the electrolyte interface. The *R*_ct_ values found in different concentrations of viruses were compared with their corresponding control values before addition of the virus and are plotted in Figure 2b as the calibration curve. The control value designated in the plot represents the charge transfer between the bare sensor electrode and electrolyte before adding any analyte, which is assigned as 100%. After virus loading, a large number of nonconducting virus molecules covered the conducting surface of N,S-GQDs, and AuNP-PAni, increasing the *R*_ct_. The change in *R*_ct_ was calculated in percentages to obtain the calibration lines. As shown in Figure 2b, the calibration line shows an excellent linear relationship between *R*_ct_ and the CHIKV concentration. The limit of detection (LOD), determined by 3*σ*/*S* (where *σ* is the standard deviation of the lowest signal and *S* is the slope of the calibration line) [25], was found to be 22.1 fg·mL^−1^. 

The main goal of this study was to detect different mosquito-borne viruses. These viruses also have similar surface functionalities that can affect the sensor specificity; thus, it was imperative to investigate the sensor’s selectivity, especially with ZIKV and DENV. To analyze the selectivity, the sensor electrode was tested with 10 pg mL^−1^ CHIKV in addition to other samples of ZIKV, DENV-LP-2, Influenza virus A/H1N1, and NoV-LP (all are 10 pg·mL^−1^). In their Nyquist plots in Figure 2c, it can be seen that the sensor responses to the other viruses are significantly lower due to the nonspecific interaction with the antibody (mentioned in a bracket of Figure 2c), indicating the specificity for the target CHIKV. The *R*_ct_ values are deciphered from the same circuit diagram, and the percentage change has been plotted in the bar diagram in Figure 2d, where the specificity can be visible.

### 3.3. Detection of ZIKV by Au|PAni|Au-PAni-N,S-GQD-Ab_ZIKV_ Sensor Electrode

The detection mechanism was based on antibody–antigen interactions; therefore, it can be anticipated that the sensor should be applicable for the detection of other virused by changing the corresponding antibodies on the sensor surface. Therefore, a similar sensor was developed for Zika virus sensing with anti-Zika antibody-conjugated Au-PAni-N,S-GQD nanocomposites. The Nyquist plot and the calibration line show identical results for CHIKV, as presented in Figure 3a. The *R*_ct_ values for each electrode were deciphered with the same circuit, and the calibration line has been drawn against concentration in Figure 3b. The LODs were also found from the calibration lines, and were as low as 31.1 fg·mL^−1^, confirming the detection mechanism of EIS-based virus sensing. Similarly, the Au|PAni|Au-PAni-N,S-GQD-Ab_ZIKV_ sensor electrode should possess high selectivity, as shown in the corresponding selectivity values in Figure 3c. 

### 3.4. Detection of Dengue Serotypes

Among the mosquito-borne virus diseases, the fatality rate of DENV infection is very high. Treatment becomes critical when secondary infection occurs by a virus serotype different from that of the initial infection [43]. Therefore, serotype identification is equally important as virus detection in the case of DENV. After successfully detecting CHIKV and ZIKV by their corresponding antibody-conjugated Au|PAni|Au-PAni-N,S-GQD sensor electrodes, this proposed method was introduced to detect four serotypes of DENV-LP, separately, for serotyping. The serotypes have many similarities; thus, there was a high possibility for cross-reactivity through the antibody. Therefore, each sensor electrode contained the corresponding monoclonal antibody for serotype sensing. It is clear from Figure 4a–d that the sensor exhibited a significant increment of impedance, increasing with concentration in their Nyquist plots for all the serotypes. Although the increments of the impendence were not of the same magnitude for all serotypes, the trends were very similar for all. The antibody–antigen interactions for different DENV serotypes are not very similar; therefore, it is obvious that virus accumulation on the electrode surface was also slightly different, although treatment with the same concentration of DENV-LP samples resulted in some variation in the impedance value. However, increasing the trend in impedance resulted in a similar concentration response for all serotypes. After measuring their *R*_ct_ values for the calibration lines in Figure 4e, the trends could be accurately expressed. It can be noted that the slopes of the all-serotypes calibration plots are very similar, which represents their corresponding LOD values of 27.4, 24.5, 41.4, and 13.3 fg mL^−1^ for DENV-LP-1, DENV-LP-2, DENV-LP-3, and DENV-LP-4, respectively.

### 3.5. Performance for Cross-Reactivity of Dengue Serotypes

Although the antibodies used for the sensor preparation were monoclonal, the chances of cross-reactivity were high due to the similarity of viral origin. To check the ability of this sensor to distinguish the identity of different serotypes, four similar electrodes with different antibody-conjugation were applied to verify the cross-reactivity of all four dengue serotypes. The interactions between the four types of sensor electrodes with four serotypes of DENV-LP and four controls are presented in Figure 5. The responses have been converted into the percentage change in *R*_ct_ for clear understanding. It can be observed from Figure 5a–d that the assigned sensor electrode, which was designed for a specific DENV-LP serotype, was very sensitive for the exact target DENV-LP serotype, showing reduced responses for other serotypes as well as the controls. Control responses for all four sensors were almost insignificant compared with other signals (near 100% for all cases). However, unlike the previous viruses of CHIKV and ZIKV, in these cases, the selectivity showed some positive responses due to the cross-reactive nature of the antibodies. In the Au|PAni|Au-PAni-N,S-GQD-Ab_DENV-2_ sensor electrode, the cross-reactivity showed a 350% enhancement in DENV-LP-1 and four of the target DENV-LP-2, where the actual target of DENV-LP-2 showed a 605% enhancement. The specificity of the antibody for the DENV-LP-2 target was observed to be the most interfering in nature; therefore, the high cross-reactivity of the sensor electrode can be justified [44,45,46]. These results also proved the successful performances of the sensor electrode, which are highly dependent on the nature of the antibody. However, due to the presence of the antibody, the sensor can possess high specificity but suffers low stability (Appendix A). It is recommended to apply the sensor electrode 1 week within its preparation due to the low stability of the antibody. 

## 4. Discussion

This paper proposes an electrochemical biosensor with PAni|Au-PAni-N,S-GQD nanocomposites, combining different antibodies to detect their corresponding viruses. The AuNP-PAni nanocomposites that conjugated N,S-GQDs-Ab enhanced the electron transfer process, which improved the electrochemical response and provided a small resistive value in the electrochemical impedance spectroscopy. The Au nanoparticles (AuNPs) with well-defined and controlled shapes incorporated in the nanochain of a conducting polymer, polyaniline, have attracted increasing attention as a promising material for biosensing matrixes. On the other hand, N,S-GQDs with structural defects in N and S can present useful functionality for nanocomposite formation. In N,S-GQDs, the nitrogen atom drastically enhances the electrochemical properties of GQD. In contrast, sulfur can increase the coordinate binding with the AuNP, situated in the Au-PAni nanocomposites. The N,S-GQDs can conjugate with the antibody embedded with Au-PAni. Therefore, the Ab-N,S-GQDs@AuNP-PAni nanocomposites show excellent electroactivity in solution, and can be applied for impedimetric virus detection. After the addition of the virus, the sensor electrode can bind with the viruses due to the conjugated antibody on the electrode surface, where the *R*_ct_ value between the sensor electrode and the electrolyte solution exhibits a significant change, resulting in a large increase in the EIS result. Due to the usage of antibody–antigen interactions on the nanocomposites, the sensor shows excellent selectivity and minimal cross-reactivity in the presence of other viruses. In the case of DENV serotyping, where the possibility of cross-reactivity is very high, this sensor can identify the serotype in the concentration of pg mL^−1^, which to the best of our knowledge, is the first attempt using an electrochemical process. In recent studies, the identification of serotypes has not been investigated in DENV detection. As listed in Table 1, most reports mainly focus on DENV detection through the NS1 protein or secondary antibody concentrations [47,48]. However, it is always better to detect the direct presence of virus concentrations rather than proteins or IgG or IgM, because their concentrations in the initial stage of infection are significantly low compared to direct virus loading. In serotyping, few reports have been published targeting the oligomers of different serotypes where the extraction of the RNA is a time-consuming process. Observing the overall performance, although the cross-reactivity was not negligible, we could still confirm qualitative information about the DENV serotype, which is highly necessary for DENV diagnosis.

## 5. Conclusions

During the past few decades, several investigations have been carried out to establish a sensitive detection technique of viruses. Although few biosensors have improved virus detection in selectivity, sensitivity, and response time, practical usages are limited, especially in cases where the analytes derive from similar origins. This report has successfully developed an electrochemical biosensor with PAni|Au-PAni-N,S-GQD nanocomposites, combining different antibodies to detect their corresponding viruses. We have targeted the mosquito-borne viruses DENV-1, DENV-2, DENV-3, DENV-4, CHIKV, and ZIKV in their detection, conjugating their corresponding antibodies on the nanocomposites. In all cases, we achieved high sensitivity, with LOD values of 22.1, 31.1, 27.4, 24.5, 41.4, and 13.3 fg mL^−1^ for CHIKV, ZIKV, DENV-LP-1, DENV-LP-2, DENV-LP-3, and DENV-LP-4, respectively. We hope that the proper development of this method for applications in disposable and multiplex systems can result in a single sensor to detect several closely related viruses in the near future.

## Figures and Tables

**Figure 1 biosensors-11-00376-f001:**
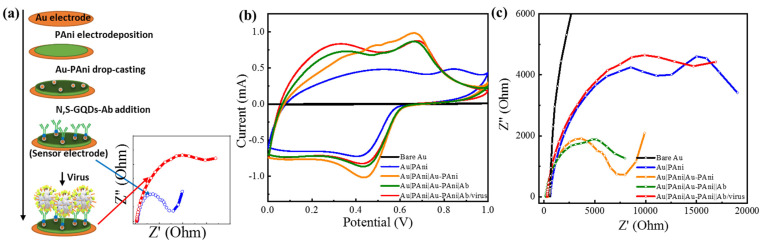
(**a**) Schematic diagram of the stepwise preparation of the sensor electrode, (**b**) cyclic voltammetry, and (**c**) electrochemical impedance results of Au|PAni|Au-PAni-Ab electrode before and after virus addition.

**Figure 2 biosensors-11-00376-f002:**
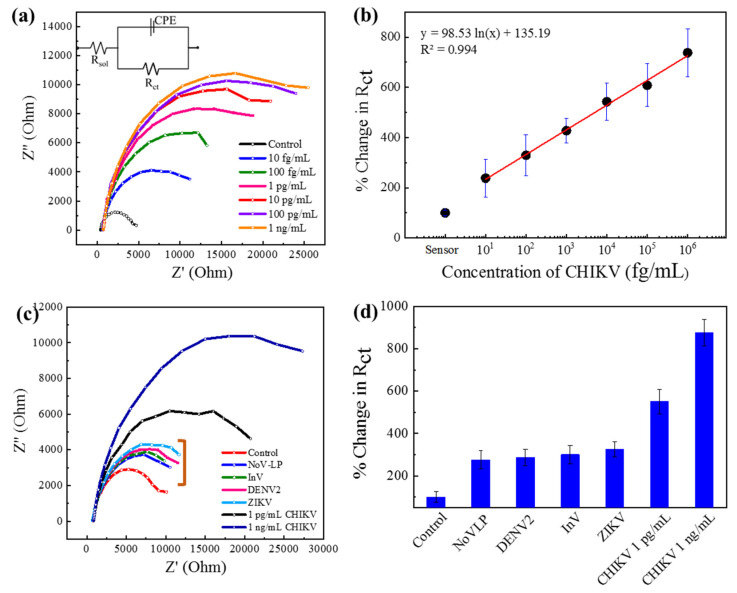
Au|PAni|Au-PAni-N,S-GQD-Ab_CHIKV_ sensor performances in different concentrations of CHKV analytes (10 fg mL^−1^–1 ng mL^−1^). (**a**) Nyquist plots (inset: equivalent circuit), (**b**) calibration line for percentage change in *R*_ct_ vs. CHIKV concentration; selectivity of the proposed sensor with 10 pg mL^−1^ of NoV-LP, DENV-LP-2, Influenzavirus A (H1N1), and ZIKV along with 1 pg mL^−1^ and 1 ng mL^−1^ target CHIKV in (**c**) Nyquist plot and (**d**) bar diagram. Error bars represent the standard deviations of triple measurements.

**Figure 3 biosensors-11-00376-f003:**
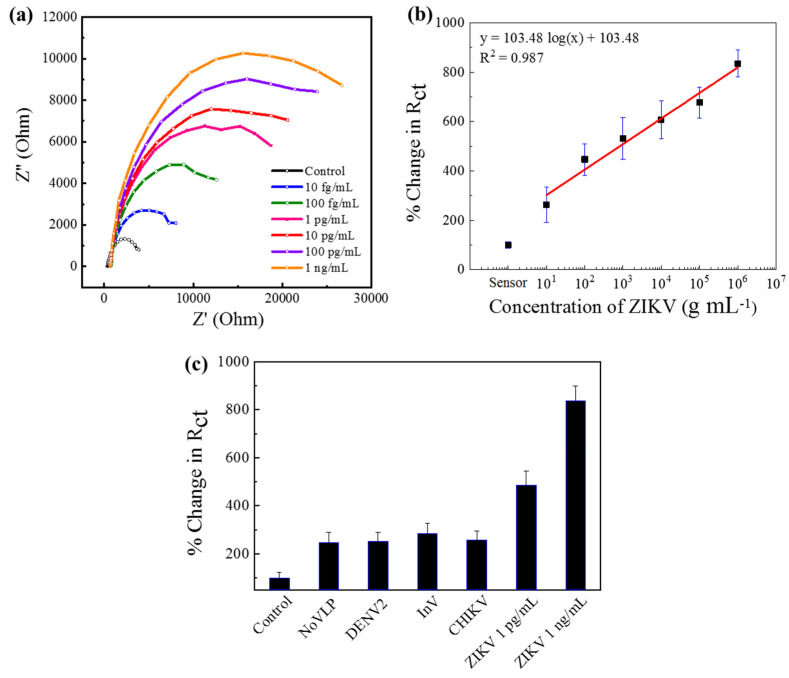
Au|PAni|Au-PAni-N,S-GQD-Ab_ZIKV_ sensor performances in different concentrations of ZIKV analytes (10 fg mL^−1^–1 ng mL^−1^). (**a**) Nyquist plots, (**b**) calibration line for percentage change in *R*_ct_ vs. ZIKV concentration, (**c**) selectivity of the proposed sensor with 10 pg mL^−1^ of NoV-LP, DENV-LP-2, Influenza virus A (H1N1), and ZIKV along with 1 pg mL^−1^, and 1 ng mL^−1^ target CHIKV. Error bars represent the standard deviations of triple measurements.

**Figure 4 biosensors-11-00376-f004:**
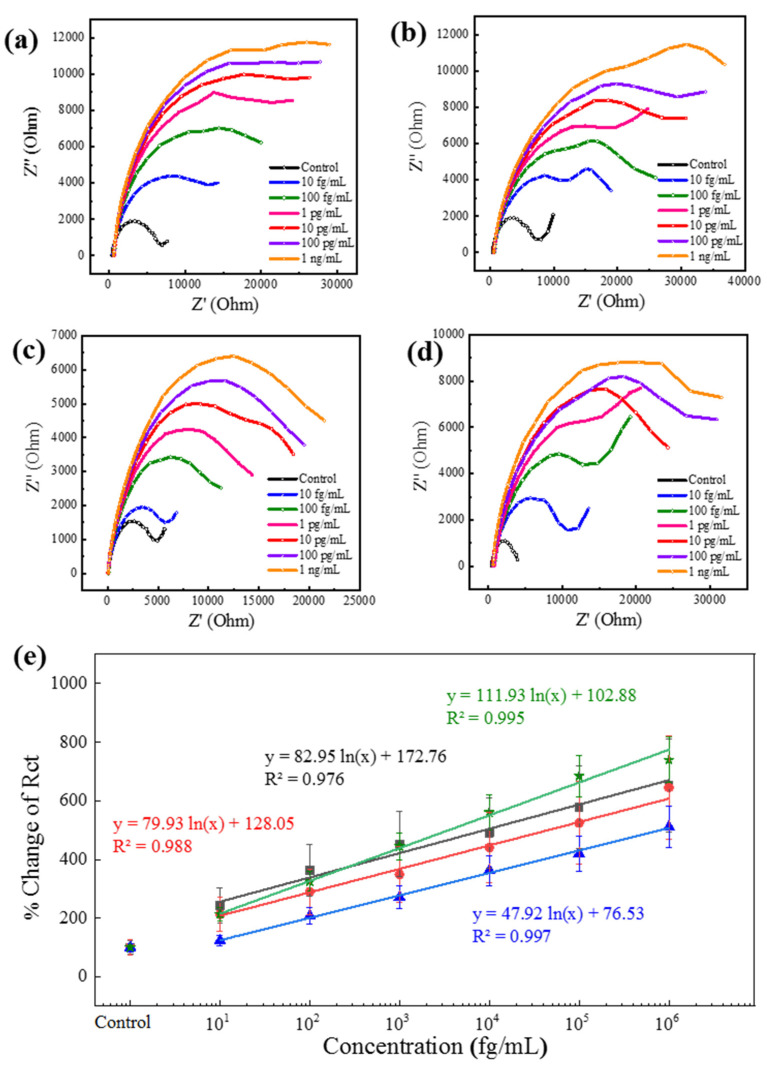
(**a**–**d**) Nyquist plots of four different DENV-LP serotype antibody-conjugated Au|PAni|Au-PAni-N,S-GQD-Ab_DENV_ electrodes for their corresponding targets in a concentration range from 100 fg mL^−1^ to 1 ng mL^−1^. (**e**) Calibration lines for all four serotypes in terms of percentage change in *R*_ct_ vs. DENV-LP concentration. Symbols: squares for DENV-LP-1; circles for DENV-LP-2; stars for DENV-LP-3; and triangles for DENV-LP-4. Error bars represent the standard deviations of triple measurements.

**Figure 5 biosensors-11-00376-f005:**
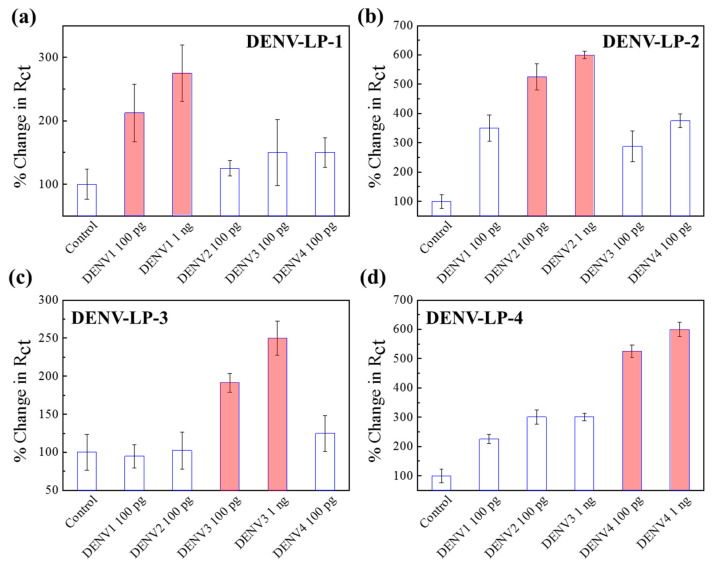
(**a**–**d**) Selectivity of four different DENV sensor electrodes with their corresponding target concentrations of 100 pg mL^−1^ and 1 ng mL^−1^ in the presence of other serotypes. Error bars represent the standard deviations of triple measurements.

**Table 1 biosensors-11-00376-t001:** Comparison of the detectability of the proposed sensor with recently reported DENV sensors.

Detection Method	Analytes	LOD	Detection Range	References
Electrochemical	DENV NS 1 protein	1.49 μg mL^−1^	0–1.4 µg mL^−1^	[49]
SPR—optical	DENV type 2 E proteins	0.08 pM	0.08–0.5 pM	[50]
Optical DNA biosensor	DENV serotype 2	10^−21^ M	1.0 × 10^−15^–1.0 × 10^−11^ M	[51]
SRP—biosensor	DENV serotype 2 and 3	2 × 10^4^ particles mL^−1^	–	[52]
Colorimetric	Different DENV serotype	–	–	[48]
Fluorometric	DENV all serotypes	9.4 fM	10^−14^ to 10^−6^ M	[25]
SERS-based lateral flow biosensor	DENV NS 1 protein	15 ng mL^−1^	15–500 ng mL^−1^	[53]
Electrochemical	DENV NS 1 protein	0.3 ng mL^−1^	1–200 ng mL^−1^	[54]
Electrochemical	DENV-LP 1 serotype	27.4 fg mL^−1^	100 fg^−1^ ng mL^−1^	This work
Electrochemical	DENV-LP 2 serotype	24.5 fg mL^−1^	100 fg^−1^ ng mL^−1^	This work
Electrochemical	DENV-LP 3 serotype	41.4 fg mL^−1^	100 fg^−1^ ng mL^−1^	This work
Electrochemical	DENV-LP 4 serotype	13.3 fg mL^−1^	100 fg^−1^ ng mL^−1^	This work

## Data Availability

The supporting data for this study are available from the corresponding author upon reasonable request.

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
