# Peer review of "Design and Analysis of a Single System of Impedimetric Biosensors for the Detection of Mosquito-Borne Viruses"

_biosensors, 2021, doi:10.3390/bios11100376_

Round 1
Reviewer 1 Report
The authors successfully designed and analysis of a single system of impedimetric biosensor for the detection of mosquito-borne viruses. Although being very interesting, I found that there are some major issues with the paper that require addressing prior to this being considered for publication in this journal. I have identified the main points for consideration below:
1. The manuscript has some styles errors, spelling typos, and grammar errors. Please carefully correct them in the revised manuscript.
2. In the introduction, the advantage of electrochemical sensors should be described, and some recent references should be cited in this section, such as Analytica Chimica Acta, 2021, 1170, 338480; Materials Science & Engineering C 109 (2020) 110615; TrAC Trends in Analytical Chemistry, 2020, 133, 116073.
3. A scheme illustration is recommended to be added in order to better show the working principle of this biosensor.
4. Reagents and biomaterials should be numbered 2.1.
5. The error bar in the Fig. 2b and Fig.3b is too big.
6. The equivalent circuit model for EIS fitting should be introduced.
7. The repeatability, reproducibility and stability of the proposed sensors should be added in the revised manuscript.
8. The real sample detection is missing.
Author Response
Biosensors (Manuscript ID biosensors-1393282)
Reviewer’s comments:
Reviewer #1:
The authors successfully designed and analysis of a single system of impedimetric biosensor for the detection of mosquito-borne viruses. Although being very interesting, I found that there are some major issues with the paper that require addressing prior to this being considered for publication in this journal. I have identified the main points for consideration below:
- The manuscript has some styles errors, spelling typos, and grammar errors. Please carefully correct them in the revised manuscript.
R: Thank you for the comment. We sincerely apologies for the mistakes. We have revised the manuscript and made corrections as per your kind suggestion.
- In the introduction, the advantage of electrochemical sensors should be described, and some recent references should be cited in this section, such as Analytica Chimica Acta, 2021, 1170, 338480; Materials Science & Engineering C 109 (2020) 110615; TrAC Trends in Analytical Chemistry, 2020, 133, 116073.
R: Thank you for raising this point. We have revised the manuscript's introduction part (L72–78 of revised MS), incorporating additional discussion with the mentioned reference list (Ref. list No. 28–30).
- A scheme illustration is recommended to be added in order to better show the working principle of this biosensor.
R: Thank you for the comment. We understand the lack of proper understanding of the principle of this biosensor; therefore, we have revised the schematic illustration of Fig. 1a according to the reviewer’s suggestion. The schematic preparation of the nanocomposite was already given in the supplementary information.
- Reagents and biomaterials should be numbered 2.1.
R: Thank you for noticing the point. We have revised the manuscript and made corrections as per the reviewer’s indication.
- The error bar in the Fig. 2b and Fig.3b is too big.
R: Thank you for the reviewer’s comment. The data points of each measurement are calculated based on at least five experiments, making the error bar (standard deviation) more oversized. However, the correlation coefficients of the calibration lines of Figs. 2b and 3b were 0.994 and 0.987, respectively, indicating good linearity between virus concentration and impedances. To represents the actual scenario in the electrochemical process, we think it is reasonable to show the full diagram.
- The equivalent circuit model for EIS fitting should be introduced.
R: Thank you for the reviewer’s suggestion. The circuit model was given in the supplementary material. However, for better representation, we have inserted in the inset of Figure 2a.
Figure 2. Au|PAni|Au-PAni-N,S-GQD-AbCHIKV sensor performances in different concentrations of CHKV analytes (10 fg mL− 1 – 1 ng mL−1). (a) Nyquist plots (inset: equivalent circuit) (b) Calibration line for percentage change of Rct vs. CHIKV concentration; selectivity of the proposed sensor with 10 pg mL–1 of NoV-LP, DENV-LP-2, Influenza virus A (H1N1) and ZIKV along with 1 pg mL–1 and 1 ng mL–1 target CHIKV in (c) Nyquist and (d) Bar diagram. Error bars represent the standard deviation of triple measurements.
- The repeatability, reproducibility and stability of the proposed sensors should be added in the revised manuscript.
R: Thank you for your valuable suggestion. All detection experiments have been carried out at least three times or more to confirm the reproducibility and repeatability. The standard deviations are also given in their corresponding calibration lines.
In case of stability, the sensor should not possess a high life-time as it contains the antibody. Though the antibody-conjugated electrodes are always kept at 4℃, however, there is still a degradation of the antibody’s performance over time. To confirm this, we have added a stability comparison over 2 weeks which strongly recommends using the sensor electrode within 1 week of its preparation.
Figure S4. Stability test of the sensor electrodes.
L319–322
These results also proved the successful performances of the sensor electrode, which are highly dependent on the nature of the antibody. However, due to the presence of antibody, the sensor can possess high specificity but suffers low stability (Figure S4). It is recommended to apply the sensor electrode 1 week within its preparation due to the low stability of the antibody.
- The real sample detection is missing.
R: Thank you for the suggestion. This is always the ultimate target of every biosensor to work on real samples. However, in this present work, we are not in a stage of real sample analysis. To proceed to the final stage, we must get ethical permission from the hospital and University, which takes a long process and more than half a year. More important things are to optimize the current work protocol towards the real sample analysis; there are many other factors that we have to overcome not to spread viruses in environment. Therefore, in this work, we only tried to establish a way of electrochemical detection of multiple viruses. In contrast, the real sample analysis in a single platform is our next step of this work, which can be done in the future.

Reviewer 2 Report
Title: Design and analysis of a single system of impedimetric biosensor for the detection of mosquito-borne viruses
Manuscript No: biosensors-1393282
Comments to authors:
The work “Design and analysis of a single system of impedimetric biosensor for the detection of mosquito-borne viruses”, is interesting and the authors have reported the most required virus detection sensors for the medical sector. I appreciate the author’s efforts and recommend publishing, after minor revision.
- The electrochemical parameters of the sensor electrode from impededance equivalent electrical circuit is important and need to be included within the main manuscript. Also if u some data on capacitance and charge transfer resistance values will be helpeful
- The authors have reported the LOD and sensing performance of the fabricated electrode clearly. However, the role and sensing performance of Au, Au-PAni and Au-PAni-N,S-GQD must be clearly mentioned in the abstract and in the discussions. Also, the choice of selecting N,S-GQDs for sensor electrode must be precisely mentioned.
- Apart from TEM images, structural and compositional characteristics for N,S-GQDs must be included for superior understanding and complete analysis of the prepared samples.
- The preparation of Au-PAni nanocomposite has to be discussed elaborately. The mechanism and process of interfacial polymerization has to be discussed.
- Is the chemicals used in the sensor are non-toxic, as the aim of the manuscript is based on virus detection and perhaps, it’s mostly employed in medical sector. Does the author have analyzed any toxicity for N,S-GQDs. Literatures reported regarding the non-toxicity or minimum toxicity of the used chemicals should be cited.
- Please check for the grammar and typos in the manuscript.
- Present paper is on applications of EIS : It will nice to mention and add few lines importance of EIS in electroanalytical techniques for energy storage and conversion etc, solid state electrolytes etc Electrochimica Acta 161(2015) 261, ACS Applied Materials and Interfaces (16)(2013)7777
Author Response
Biosensors (Manuscript ID biosensors-1393282)
Reviewer’s comments:
Reviewer #2
The work “Design and analysis of a single system of impedimetric biosensor for the detection of mosquito-borne viruses”, is interesting and the authors have reported the most required virus detection sensors for the medical sector. I appreciate the author’s efforts and recommend publishing, after minor revision.
R: Thank you very much for appreciating our work. It will give us more inspiration to think and work in a better way.
- The electrochemical parameters of the sensor electrode from impedance equivalent electrical circuit is important and need to be included within the main manuscript. Also if u some data on capacitance and charge transfer resistance values will be helpful.
R: Thank you for the comment. According to the reviewer’s suggestion, we put the circuit diagram of the impedance circuit in the main manuscript and also put the values of the Rct and C in a table of supplementary information Table S1-a and S1-b.
- The authors have reported the LOD and sensing performance of the fabricated electrode clearly. However, the role and sensing performance of Au, Au-PAni and Au-PAni-N,S-GQD must be clearly mentioned in the abstract and in the discussions. Also, the choice of selecting N,S-GQDs for sensor electrode must be precisely mentioned.
R: Thank you very much for raising this point. We have revised the introduction and the discussion of the manuscript, mentioning the role of individual parameters according to the reviewer’s suggestion. However, due to the size, we avoid to introduce about these in the abstract.
L74–78,
“In this work, the nanocomposites of Au-PAni and N,S-GQDs are conjugated together with different antibodies and thereafter applied for their corresponding target viruses by the im-pedimetric process. The plenty of carboxylic groups on GQDs can covalently be attached with the free amino group of antibodies, which makes the electrode surface stable and specific for detection.”
L340–350,
“The Au nanoparticles (AuNPs) with well-defined and controlled shapes incorporated in the nanochain of conducting polymer, polyaniline, have attracted increasing attention as a promising material for biosensing matrix. On the other hand, N,S-GQDs with the structural defects with N and S, can generate useful functionality for nanocomposite formation. In N,S-GQDs, the nitrogen atom could drastically enhance the electrochemical properties of GQD. In contrast, the sulfur can increase the coordinate binding with the AuNP, situated in the Au-PAni nanocomposites. The N,S-GQDs can conjugate with the antibody, which is embedded with Au-PAni. Therefore the Ab-N,S-GQDs@AuNP-PAni nanocomposites show excellent electroactivity in solution, which can be applied for the impedimetric virus detection.”
- Apart from TEM images, structural and compositional characteristics for N,S-GQDs must be included for superior understanding and complete analysis of the prepared samples.
R: Thank you for the comment. The N,S-GQDs are already characterized in our previous study which is cited in this work where the HRTEM, fringes calculation, absorption, fluorescence, XRD, Raman etc. all tools supported the successful preparation of the N,S-GQDs. In this work, the same N,S-GQDs are used for the conjugation. Therefore, we have mentioned the earlier citation in this work for GQDs characterization.
- The preparation of Au-PAni nanocomposite has to be discussed elaborately. The mechanism and process of interfacial polymerization has to be discussed.
R: Thank you for the reviewer’s comment. We have added few lines about the interfacial synthesis of the Au-PAni nanocomposites in the revised manuscript.
L130–134,
“AuNP-PAni nanocomposites were synthesized via interfacial self-oxidation-reduction polymerization with HAuCl4 as aqueous oxidant and the poly-aniline as a monomer in the organic toluene layer [35]. These two immiscible layers contact in their interfaces, and then the Au3+ ions oxidize the aniline monomer to its conducting emeraldine salt polymer formation in the nanotube structure, whereas itself reduced to the nano Au0 form. The AuNP is therefore entrapped on the nanotube surface of the Polyaniline, resulting AuNP-PAni nanocomposites. Finally, the AuNP-PAni nanocomposites were drop-casted on the PAni-coated Au electrode for further analysis.”
- Is the chemicals used in the sensor are non-toxic, as the aim of the manuscript is based on virus detection and perhaps, it’s mostly employed in medical sector. Does the author have analyzed any toxicity for N,S-GQDs. Literatures reported regarding the non-toxicity or minimum toxicity of the used chemicals should be cited.
R: Thank you for raising this point. N,S-GQD is a well-known and widely used fluorescent QDs in biosensing field. The N,S-GQD amount, used in this work is very less in concentration which is insignificant to the environment though it seems to be a non-toxic or less toxic towards environment. Due to their promising characteristics and easy fabrication procedure, these GQDs are very reliable for use in establishing a new sensing system, here. In addition, there are no possibility to use this sensor in vivo application, the cytotoxicity measurement of the sensor can be avoided.
- Please check for the grammar and typos in the manuscript.
R: Thank you for the comment. We sincerely apologies for the mistakes. We have revised the manuscript and made corrections thoroughly. After that, we have also requested one of native speaker friend to check the grammar and sentence construction.
- Present paper is on applications of EIS: It will nice to mention and add few lines importance of EIS in electroanalytical techniques for energy storage and conversion etc, solid state electrolytes etc Electrochimica Acta 161(2015) 261, ACS Applied Materials and Interfaces (16)(2013)7777.
R: Thank you very much for your kind suggestion. We have revised the manuscript according to the reviewer’s comment and the above-mentioned references have been cited in the revised manuscript (No. 32 and 33).

Reviewer 3 Report
The article biosensors-1393282 entitled “Design and analysis of a single system of impedimetric biosensor for the detection of mosquito-borne viruses” described the preparation of different immunosensors based on the use of gold electrodes covered by a thin PANI film in which a nanocomposite material based on PANI nanotubes decorated with gold nanoparticles are deposited. After that graphene nanosheets previously modified with the specific capture antibody of different viruses (dengue virus (DENV), zika virus 12 (ZIKV), and chikungunya virus (CHIKV)) are immobilized over the electrode surface. These modified electrodes have been used as an impedimetric sensor, based on the accumulation of the different virus capsids, characterized by their non-electrical conductive, generating the increase of semicircles radius in the Nyquist Plots. Unless the paper is well write and the results are interesting, it must to be mention that similar examples have been described previously in bibliography:
- Sensors and Actuators B 220 (2015) 565-572 http://dx.doi.org/10.1016/j.snb.2015.05.067
- Dutta R, Thangapandi K, Mondal S, et al. Polyaniline Based Electrochemical Sensor for the Detection of Dengue Virus Infection. Avicenna J Med Biotechnol. 2020;12(2):77-84.
I recommend citing these articles as they are really close to the evaluated one.
Despite of that, the good analytical parameters showed in this paper, improving the describes figures of merit of other paper, made me consider it as a good work to be published in biosensor.
Only a few recommendations to be considered:
- I miss some electrode surface characterization: SEM, XPS, RAMAN, AFM… of the different steps of immunosensor fabrication.
- The chemisorption of sulphur atom included in the N,S-GQD onto gold nanoparticles I consider is something to discuss, authors should try to prove it with some technique or citing some previous work that has probed this event. It is widely knows the adsorption of thiols over gold surfaces, but I have never have concerned that N,S-GQD can also by chemisorbed over gold. Just probe this or cite a proper paper that probe it.
Author Response
Biosensors (Manuscript ID biosensors-1393282)
Reviewer’s comments:
Reviewer #3:
The article biosensors-1393282 entitled “Design and analysis of a single system of impedimetric biosensor for the detection of mosquito-borne viruses” described the preparation of different immunosensors based on the use of gold electrodes covered by a thin PANI film in which a nanocomposite material based on PANI nanotubes decorated with gold nanoparticles are deposited. After that graphene nanosheets previously modified with the specific capture antibody of different viruses (dengue virus (DENV), zika virus 12 (ZIKV), and chikungunya virus (CHIKV)) are immobilized over the electrode surface. These modified electrodes have been used as an impedimetric sensor, based on the accumulation of the different virus capsids, characterized by their non-electrical conductive, generating the increase of semicircles radius in the Nyquist Plots. Unless the paper is well write and the results are interesting, it must to be mention that similar examples have been described previously in bibliography:
- Sensors and Actuators B 220 (2015) 565-572 http://dx.doi.org/10.1016/j.snb.2015.05.067
- Dutta R, Thangapandi K, Mondal S, et al. Polyaniline Based Electrochemical Sensor for the Detection of Dengue Virus Infection. Avicenna J Med Biotechnol. 2020;12(2):77-84.
I recommend citing these articles as they are really close to the evaluated one.
R: Thank you very much for your kind suggestion. We have revised the manuscript according to the reviewer’s comment along with the reference list. The above-mentioned references have been cited in the revised manuscript (Ref. List No. 44 and 52).
Despite of that, the good analytical parameters showed in this paper, improving the describes figures of merit of other paper, made me consider it as a good work to be published in biosensor.
R: Thank you very much for appreciating our work. It will give us more inspiration to think and work in a better way.
Only a few recommendations to be considered:
- I miss some electrode surface characterization: SEM, XPS, RAMAN, AFM… of the different steps of immunosensor fabrication.
R: Thank you for the comment. The characterization of each consecutive step of electrode fabrication is a valuable experiment. However, analyzing the sensor electrode in any type of surface characterization can disrupt its morphology. So, we can only characterize by electrochemical CV. In the case of materials characterizations instead of the electrode, that should be possible. However, the primary materials of this sensor electrode like Au-PAni and N,S-GQDs have already been used for our previous electrochemical analysis, which we have cited here. To make this work more concise, we prefer to cite our previous but recent publications for detailed characterizations and focused here only on the sensing part.
- The chemisorption of sulphur atom included in the N,S-GQD onto gold nanoparticles I consider is something to discuss, authors should try to prove it with some technique or citing some previous work that has probed this event. It is widely knows the adsorption of thiols over gold surfaces, but I have never have concerned that N,S-GQD can also by chemisorbed over gold. Just prove this or cite a proper paper that prove it.
R: Thank you for raising this point. The coordination bonding between sulfur and the AuNP has been used widely. In addition, in some recent reports, this bonding has already been established on S-doped GQDs and AuNP. One of our previous papers also reported the same where they characterized these by XPS and elemental mapping. So, in this revised manuscript, we have mentioned those reports as references.

Round 2
Reviewer 1 Report
Accept in present form.
Author Response

(The authors gave the same response as above.)

Reviewer 3 Report
In my opinion, authors do not answer properly one of the main question I proposed during the first revision round:
- The chemisorption of sulphur atom included in the N,S-GQD onto gold nanoparticles I consider is something to discuss, authors should try to prove it with some technique or citing some previous work that has probed this event. It is widely knows the adsorption of thiols over gold surfaces, but I have never have concerned that N,S-GQD can also by chemisorbed over gold. Just probe this or cite a proper paper that probe it.
I think they do not prove it directly through experiment, and they do not cite the proper bibliography in order to support this important point, as it is fundamental for understand the immunosensor way of work.
Author Response
Biosensors (Manuscript ID biosensors-1393282R1)
Reviewer’s comments:
Reviewer #3:
In my opinion, authors do not answer properly one of the main question I proposed during the first revision round:
- The chemisorption of sulphur atom included in the N,S-GQD onto gold nanoparticles I consider is something to discuss, authors should try to prove it with some technique or citing some previous work that has probed this event. It is widely knows the adsorption of thiols over gold surfaces, but I have never have concerned that N,S-GQD can also by chemisorbed over gold. Just probe this or cite a proper paper that probe it.
I think they do not prove it directly through experiment, and they do not cite the proper bibliography in order to support this important point, as it is fundamental for understand the immunosensor way of work.
Answer: Thank you very much for your comments. As the Au-thiol chemistry is very common in bioconjugation and sensing related work, initially, we might not be addressed the point, properly. However, according to the reviewer, the conjugation between thiolated GQD and AuNP should be properly established. In our old data of ICP-OES (Inductively Coupled Plasma-Optical Emission Spectrometry) analysis, where we estimated the amount of Sulphur in N,S-GQD and Au-PAni-N,S-GQDs, the conjugation was indirectly estimated. The high amount of Sulphur in Au-PAni-N,S-GQDs nanocomposites proves the successful conjugation which predominantly come from the Au-S bonding. As a control, we also counted the N atom in case of N-GQD and AuNP-PAni which is not as much as previous amount, as the conjugation of GQD in the AuNP is occurred due to the weak interaction. So, it is clear from the analysis that though there are some interactions between GQD with Au-PAni, however the major conjugation between these two nanocomposites is governed by the AuNP and S chemistry. The results are shown below in the Table; however, we cannot mention this table in the main manuscript as these data are taken from one of our on-going project. Therefore, we have added two important references in the manuscript where the Au-S chemistry was used in the similar purpose.
|
N,S-GQD |
S (20 ppm) |
|
|
Au-PAni-N,S-GQD |
S (3 ppm) |
|
|
N-GQD |
|
N (30 ppm) |
|
Au-PAni-N-GQD |
|
N (<1 ppm) |
L193–195
The conjugation between the sulphur atom of N,S-GQD and the AuNP of AuPAni has been made by the universal gold thiol interaction [40,41].
Cited References:
- Mahmoud, A.M.; El-Wekil, M.M.; Mahnashi, M.H.; Ali, M.F.; Alkahtani, S.A. Modification of N, S co-doped graphene quantum dots with p-aminothiophenol-functionalized gold nanoparticles for molecular imprint-based voltammetric determination of the antiviral drug sofosbuvir. Microchim. Acta 2019, 186, 617.
- Yao, J.; Li, Y.; Xie, M.; Yang, Q.; Liu, T. The electrochemical behaviors and kinetics of AuNPs/N, S-GQDs composite electrode: A novel label-free amplified BPA aptasensor with extreme sensitivity and selectivity. J. Mol. Liq. 2020, 320, 114384.
